# Lexical and Hierarchical Topic Regression

**Viet-An Nguyen**
Computer Science
University of Maryland
College Park, MD
vietan@cs.umd.edu

**Jordan Boyd-Graber**
iSchool & UMIACS
University of Maryland
College Park, MD
jbg@umiacs.umd.edu

**Philip Resnik**
Linguistics & UMIACS
University of Maryland
College Park, MD
resnik@umd.edu

## Abstract

Inspired by a two-level theory from political science that unifies agenda setting and ideological framing, we propose supervised hierarchical latent Dirichlet allocation (SHLDA), which jointly captures documents' multi-level topic structure and their polar response variables. Our model extends the nested Chinese restaurant processes to discover tree-structured topic hierarchies and uses both per-topic hierarchical and per-word lexical regression parameters to model response variables. SHLDA improves prediction on political affiliation and sentiment tasks in addition to providing insight into how topics under discussion are framed.

## 1   Introduction: Agenda Setting and Framing in Hierarchical Models

How do liberal-leaning bloggers talk about immigration in the US? What do conservative politicians have to say about education? How do Fox News and MSNBC differ in their language about the gun debate? Such questions concern not only *what*, but *how* things are talked about.

In political communication, the question of "what" falls under the heading of *agenda setting* theory, which concerns the issues introduced into political discourse (e.g., by the mass media) and their influence over public priorities [1]. The question of "how" concerns *framing*: the way the presentation of an issue reflects or encourages a particular perspective or interpretation [2]. For example, the rise of the "innocence frame" in the death penalty debate, emphasizing the irreversible consequence of mistaken convictions, has led to a sharp decline in the use of capital punishment in the US [3].

In its concern with the subjects or issues under discussion in political discourse, agenda setting maps neatly to topic modeling [4] as a means of discovering and characterizing those issues [5]. Interestingly, one line of communication theory seeks to unify agenda setting and framing by viewing frames as a second-level kind of agenda [1]: just as agenda setting is about which objects of discussion are salient, framing is about the salience of *attributes* of those objects. The key is that what communications theorists consider an attribute in a discussion can itself be an object, as well. For example, "mistaken convictions" is one attribute of the death penalty discussion, but it can also be viewed as an object of discussion in its own right.

This two-level view leads naturally to the idea of using a hierarchical topic model to formalize both agendas and frames within a uniform setting. In this paper, we introduce a new model to do exactly that. The model is predictive: it represents the idea of alternative or competing perspectives via a continuous-valued *response variable*. Although inspired by the study of political discourse, associating texts with "perspectives" is more general and has been studied in sentiment analysis, discovery of regional variation, and value-sensitive design. We show experimentally that the model's hierarchical structure improves prediction of perspective in both a political domain and on sentiment analysis tasks, and we argue that the topic hierarchies exposed by the model are indeed capturing structure in line with the theory that motivated the work.

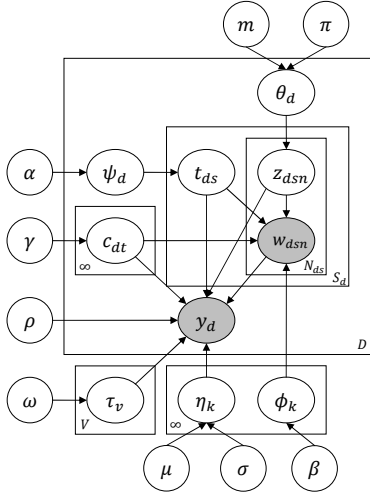

1. For each node $k \in [1, \infty)$ in the tree
   (a) Draw topic $\phi_k \sim \text{Dir}(\beta_k)$
   (b) Draw regression parameter $\eta_k \sim \mathcal{N}(\mu, \sigma)$
2. For each word $v \in [1, V]$, draw $\tau_v \sim \text{Laplace}(0, \omega)$
3. For each document $d \in [1, D]$
   (a) Draw level distribution $\theta_d \sim \text{GEM}(m, \pi)$
   (b) Draw table distribution $\psi_d \sim \text{GEM}(\alpha)$
   (c) For each table $t \in [1, \infty)$, draw a path $c_{d,t} \sim \text{nCRP}(\gamma)$
   (d) For each sentence $s \in [1, S_d]$, draw a table indicator $t_{d,s} \sim \text{Mult}(\psi_d)$
      i. For each token $n \in [1, N_{d,s}]$
         A. Draw level $z_{d,s,n} \sim \text{Mult}(\theta_d)$
         B. Draw word $w_{d,s,n} \sim \text{Mult}(\phi_{c_{d,t_{d,s}}, z_{d,s,n}})$
   (e) Draw response $y_d \sim \mathcal{N}(\boldsymbol{\eta}^T \bar{\boldsymbol{z}}_d + \boldsymbol{\tau}^T \bar{\boldsymbol{w}}_d, \rho)$:
      i. $\bar{z}_{d,k} = \frac{1}{N_{d,\cdot}} \sum_{s=1}^{S_d} \sum_{n=1}^{N_{d,s}} \mathbb{I}[k_{d,s,n} = k]$
      ii. $\bar{w}_{d,v} = \frac{1}{N_{d,\cdot}} \sum_{s=1}^{S_d} \sum_{n=1}^{N_{d,s}} \mathbb{I}[w_{d,s,n} = v]$

Figure 1: SHLDA's generative process and plate diagram. Words $\boldsymbol{w}$ are explained by topic hierarchy $\boldsymbol{\phi}$, and response variables $\boldsymbol{y}$ are explained by per-topic regression coefficients $\boldsymbol{\eta}$ and global lexical coefficients $\boldsymbol{\tau}$.

## 2   SHLDA: Combining Supervision and Hierarchical Topic Structure

Jointly capturing supervision and hierarchical topic structure falls under a class of models called *supervised hierarchical latent Dirichlet allocation*. These models take as input a set of $D$ documents, each of which is associated with a response variable $y_d$, and output a hierarchy of topics which is informed by $y_d$. Zhang et al. [6] introduce the SHLDA family, focusing on a categorical response. In contrast, our novel model (which we call SHLDA for brevity), uses continuous responses. At its core, SHLDA's document generative process resembles a combination of hierarchical latent Dirichlet allocation [7, HLDA] and the hierarchical Dirichlet process [8, HDP]. HLDA uses the nested Chinese restaurant process (nCRP($\gamma$)), combined with an appropriate base distribution, to induce an unbounded tree-structured hierarchy of topics: general topics at the top, specific at the bottom. A document is generated by traversing this tree, at each level creating a new child (hence a new path) with probability proportional to $\gamma$ or otherwise respecting the "rich-get-richer" property of a CRP.

A drawback of HLDA, however, is that each document is restricted to only a *single path* in the tree. Recent work relaxes this restriction through different priors: nested HDP [9], nested Chinese franchises [10] or recursive CRPs [11]. In this paper, we address this problem by allowing documents to have *multiple paths* through the tree by leveraging information at the sentence level using the two-level structure used in HDP. More specifically, in the HDP's Chinese restaurant franchise metaphor, customers (i.e., tokens) are grouped by sitting at tables and each table takes a dish (i.e., topic) from a *flat* global menu. In our SHLDA, dishes are organized in a *tree-structured* global menu by using the nCRP as prior. Each path in the tree is a collection of $L$ dishes (one for each level) and is called a *combo*. SHLDA groups sentences of a document by assigning them to tables and associates each table with a combo, and thus, models each document as a *distribution* over combos.[1]

In SHLDA's metaphor, customers come in a restaurant and sit at a table in groups, where each group is a sentence. A sentence $\boldsymbol{w}_{d,s}$ enters restaurant $d$ and selects a table $t$ (and its associated combo) with probability proportional to the number of sentences $S_{d,t}$ at that table; or, it sits at a new table with probability proportional to $\alpha$. After choosing the table (indexed by $t_{d,s}$), if the table is new, the group will select a combo of dishes (i.e., a path, indexed by $c_{d,t}$) from the tree menu. Once a combo is in place, each token in the sentence chooses a "level" (indexed by $z_{d,s,n}$) in the combo, which specifies the topic ($\phi_{k_{d,s,n}} \equiv \phi_{c_{d,t_{d,s}}, z_{d,s,n}}$) producing the associated observation (Figure 2).

SHLDA also draws on supervised LDA [12, SLDA] associating each document $d$ with an observable continuous response variable $y_d$ that represents the author's perspective toward a topic, e.g., positive vs. negative sentiment, conservative vs. liberal ideology, etc. This lets us infer a multi-level topic structure informed by how topics are "framed" with respect to positions along the $y_d$ continuum.

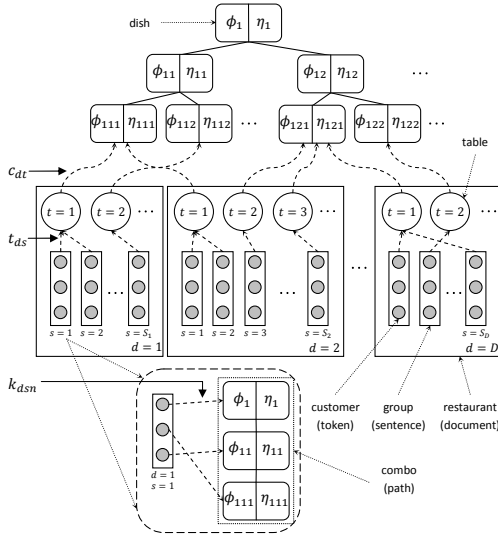

| | |
|---|---|
| $S_d$ | # sentences in document $d$ |
| $S_{d,t}$ | # groups (i.e. sentences) sitting at table $t$ in restaurant $d$ |
| $N_{d,s}$ | # tokens $\boldsymbol{w}_{d,s}$ |
| $N_{d,\cdot,l}$ | # tokens in $\boldsymbol{w}_d$ assigned to level $l$ |
| $N_{d,\cdot,>l}$ | # tokens in $\boldsymbol{w}_d$ assigned to level $> l$ |
| $N_{d,\cdot,\geq l}$ | $\equiv N_{d,\cdot,l} + N_{d,\cdot,>l}$ |
| $M_{c,l}$ | # tables at level $l$ on path $c$ |
| $C_{c,l,v}$ | # word type $v$ assigned to level $l$ on path $c$ |
| $C_{d,x,l,v}$ | # word type $v$ in $\boldsymbol{v}_{d,x}$ assigned to level $l$ |
| $\phi_k$ | Topic at node $k$ |
| $\eta_k$ | Regression parameter at node $k$ |
| $\tau_v$ | Regression parameter of word type $v$ |
| $c_{d,t}$ | Path assignment for table $t$ in restaurant $d$ |
| $t_{d,s}$ | Table assignment for group $\boldsymbol{w}_{d,s}$ |
| $z_{d,s,n}$ | Level assignment for $w_{d,s,n}$ |
| $k_{d,s,n}$ | Node assignment for $w_{d,s,n}$ (i.e., node at level $z_{d,s,n}$ on path $c_{d,t_{d,s}}$) |
| $L$ | Height of the tree |
| $\mathcal{C}^+$ | Set of all possible paths (including new ones) of the tree |

Figure 2: SHLDA's restaurant franchise metaphor.     Table 1: Notation used in this paper

Unlike SLDA, we model the response variables using a normal linear regression that contains *both* per-topic hierarchical and per-word lexical regression parameters. The *hierarchical regression parameters* are just like topics' regression parameters in SLDA: each topic $k$ (here, a tree node) has a parameter $\eta_k$, and the model uses the empirical distribution over the nodes that generated a document as the regressors. However, the hierarchy in SHLDA makes it possible to discover relationships between topics and the response variable that SLDA's simple latent space obscures. Consider, for example, a topic model trained on Congressional debates. Vanilla LDA would likely discover a *healthcare* category. SLDA [12] could discover a pro-Obamacare topic and an anti-Obamacare topic. SHLDA could do that *and* capture the fact that there are alternative perspectives, i.e., that the healthcare issue is being discussed from two ideological perspectives, along with characterizing *how* the higher level topic is discussed by those on both sides of that ideological debate.

Sometimes, of course, words are strongly associated with extremes on the response variable continuum regardless of underlying topic structure. Therefore, in addition to hierarchical regression parameters, we include global *lexical regression parameters* to model the interaction between specific words and response variables. We denote the regression parameter associated with a word type $v$ in the vocabulary as $\tau_v$, and use the normalized frequency of $v$ in the documents to be its regressor.

Including both hierarchical and lexical parameters is important. For detecting ideology in the US, "liberty" is an effective indicator of conservative speakers regardless of context; however, "cost" is a conservative-leaning indicator in discussions about environmental policy but liberal-leaning in debates about foreign policy. For sentiment, "wonderful" is globally a positive word; however, "unexpected" is a positive descriptor of books but a negative one of a car's steering. SHLDA captures these properties in a single model.

## 3   Posterior Inference and Optimization

Given documents with observed words $\boldsymbol{w} = \{w_{d,s,n}\}$ and response variables $\boldsymbol{y} = \{y_d\}$, the inference task is to find the posterior distribution over: the tree structure including topic $\phi_k$ and regression parameter $\eta_k$ for each node $k$, combo assignment $c_{d,t}$ for each table $t$ in document $d$, table assignment $t_{d,s}$ for each sentence $s$ in a document $d$, and level assignment $z_{d,s,n}$ for each token $w_{d,s,n}$. We approximate SHLDA's posterior using stochastic EM, which alternates between a Gibbs sampling E-step and an optimization M-step. More specifically, in the E-step, we integrate out $\boldsymbol{\psi}$, $\boldsymbol{\theta}$ and $\boldsymbol{\phi}$ to construct a Markov chain over $(\boldsymbol{t}, \boldsymbol{c}, \boldsymbol{z})$ and alternate sampling each of them from their conditional distributions. In the M-step, we optimize the regression parameters $\boldsymbol{\eta}$ and $\boldsymbol{\tau}$ using L-BFGS [13].

Before describing each step in detail, let us define the following probabilities. For more thorough derivations, please see the supplement.

- First, define $\boldsymbol{v}_{d,x}$ as a set of tokens (e.g., a token, a sentence or a set of sentences) in document $d$. The conditional density of $\boldsymbol{v}_{d,x}$ being assigned to path $c$ given all other assignments is

$$f_c^{-d,x}(\boldsymbol{v}_{d,x}) = \prod_{l=1}^{L} \frac{\Gamma(C_{c,l,\cdot}^{-d,x} + V\beta_l)}{\Gamma(C_{c,l,\cdot}^{-d,x} + C_{d,x,l,\cdot} + V\beta_l)} \prod_{v=1}^{V} \frac{\Gamma(C_{c,l,v}^{-d,x} + C_{d,x,l,v} + \beta_l)}{\Gamma(C_{c,l,v}^{-d,x} + \beta_l)} \qquad (1)$$

where superscript $^{-d,x}$ denotes the same count excluding assignments of $\boldsymbol{v}_{d,x}$; marginal counts are represented by $\cdot$'s. For a new path $c^{new}$, if the node does not exist, $C_{c^{new},l,v}^{-d,x} = 0$ for all word types $v$.
- Second, define the conditional density of the response variable $y_d$ of document $d$ given $\boldsymbol{v}_{d,x}$ being assigned to path $c$ and all other assignments as $g_c^{-d,x}(y_d) =$

$$\mathcal{N}\left(\frac{1}{N_{d,\cdot}}\left(\sum_{w_{d,s,n} \in \{\boldsymbol{w}_d \setminus \boldsymbol{v}_{d,x}\}} \eta_{c_{d,t_{d,s}},z_{d,s,n}} + \sum_{l=1}^{L} \eta_{c,l} \cdot C_{d,x,l,\cdot} + \sum_{s=1}^{S_d} \sum_{n=1}^{N_{d,s}} \tau_{w_{d,s,n}}\right), \rho\right) \qquad (2)$$

where $N_{d,\cdot}$ is the total number of tokens in document $d$. For a new node at level $l$ on a new path $c^{new}$, we integrate over all possible values of $\eta_{c^{new},l}$.

**Sampling $t$:** For each group $\boldsymbol{w}_{d,s}$ we need to sample a table $t_{d,s}$. The conditional distribution of a table $t$ given $\boldsymbol{w}_{d,s}$ and other assignments is proportional to the number of sentences sitting at $t$ times the probability of $\boldsymbol{w}_{d,s}$ and $y_d$ being observed under this assignment. This is $P(t_{d,s} = t \,|\, \text{rest}) \propto P(t_{d,s} = t \,|\, \boldsymbol{t}_d^{-s}) \cdot P(\boldsymbol{w}_{d,s}, y_d \,|\, t_{d,s} = t, \boldsymbol{w}^{-d,s}, \boldsymbol{t}^{-d,s}, \boldsymbol{z}, \boldsymbol{c}, \boldsymbol{\eta})$

$$\propto \begin{cases} S_{d,t}^{-d,s} \cdot f_{c_{d,t}}^{-d,s}(\boldsymbol{w}_{d,s}) \cdot g_{c_{d,t}}^{-d,s}(y_d), & \text{for existing table } t; \\ \alpha \cdot \sum_{c \in \mathcal{C}^+} P(c_{d,t^{new}} = c \,|\, \boldsymbol{c}^{-d,s}) \cdot f_c^{-d,s}(\boldsymbol{w}_{d,s}) \cdot g_c^{-d,s}(y_d), & \text{for new table } t^{new}. \end{cases} \qquad (3)$$

For a new table $t^{new}$, we need to sum over all possible paths $\mathcal{C}^+$ of the tree, including new ones. For example, the set $\mathcal{C}^+$ for the tree shown in Figure 2 consists of four existing paths (ending at one of the four leaf nodes) and three possible new paths (a new leaf off of one of the three internal nodes). The prior probability of path $c$ is: $P(c_{d,t^{new}} = c \,|\, \boldsymbol{c}^{-d,s}) \propto$

$$\begin{cases} \prod_{l=2}^{L} \frac{M_{c,l}^{-d,s}}{M_{c,l-1}^{-d,s} + \gamma_{l-1}}, & \text{for an existing path } c; \\ \frac{\gamma_{l^*}}{M_{c^{new},l^*}^{-d,s} + \gamma_{l^*}} \prod_{l=2}^{l^*} \frac{M_{c^{new},l}^{-d,s}}{M_{c^{new},l-1}^{-d,s} + \gamma_{l-1}}, & \begin{array}{l} \text{for a new path } c^{new} \text{ which consists of an existing path} \\ \text{from the root to a node at level } l^* \text{ and a new node.} \end{array} \end{cases} \qquad (4)$$

**Sampling $z$:** After assigning a sentence $\boldsymbol{w}_{d,s}$ to a table, we assign each token $w_{d,s,n}$ to a level to choose a dish from the combo. The probability of assigning $w_{d,s,n}$ to level $l$ is

$$P(z_{d,s,n} = l \,|\, \text{rest}) \propto P(z_{d,s,n} = l \,|\, \boldsymbol{z}_d^{-s,n}) P(w_{d,s,n}, y_d \,|\, z_{d,s,n} = l, \boldsymbol{w}^{-d,s,n}, \boldsymbol{z}^{-d,s,n}, \boldsymbol{t}, \boldsymbol{c}, \boldsymbol{\eta}) \quad (5)$$

The first factor captures the probability that a customer in restaurant $d$ is assigned to level $l$, conditioned on the level assignments of all other customers in restaurant $d$, and is equal to

$$P(z_{d,s,n} = l \,|\, \boldsymbol{z}_d^{-s,n}) = \frac{m\pi + N_{d,\cdot,l}^{-d,s,n}}{\pi + N_{d,\cdot,\geq l}^{-d,s,n}} \prod_{j=1}^{l-1} \frac{(1-m)\pi + N_{d,\cdot,>j}^{-d,s,n}}{\pi + N_{d,\cdot,\geq j}^{-d,s,n}},$$

The second factor is the probability of observing $w_{d,s,n}$ and $y_d$, given that $w_{d,s,n}$ is assigned to level $l$: $P(w_{d,s,n}, y_d \,|\, z_{d,s,n} = l, \boldsymbol{w}^{-d,s,n}, \boldsymbol{z}^{-d,s,n}, \boldsymbol{t}, \boldsymbol{c}, \boldsymbol{\eta}) = f_{c_{d,t_{d,s}}}^{-d,s,n}(w_{d,s,n}) \cdot g_{c_{d,t_{d,s}}}^{-d,s,n}(y_d)$.

**Sampling $c$:** After assigning customers to tables and levels, we also sample path assignments for all tables. This is important since it can change the assignments of all customers sitting at a table, which leads to a well-mixed Markov chain and faster convergence. The probability of assigning table $t$ in restaurant $d$ to a path $c$ is

$$P(c_{d,t} = c \,|\, \text{rest}) \propto P(c_{d,t} = c \,|\, \boldsymbol{c}^{-d,t}) \cdot P(\boldsymbol{w}_{d,t}, y_d \,|\, c_{d,t} = c, \boldsymbol{w}^{-d,t}, \boldsymbol{c}^{-d,t}, \boldsymbol{t}, \boldsymbol{z}, \boldsymbol{\eta}) \qquad (6)$$

where we slightly abuse the notation by using $\boldsymbol{w}_{d,t} \equiv \cup_{\{s|t_{d,s}=t\}} \boldsymbol{w}_{d,s}$ to denote the set of customers in all the groups sitting at table $t$ in restaurant $d$. The first factor is the prior probability of a path given all tables' path assignments $\boldsymbol{c}^{-d,t}$, excluding table $t$ in restaurant $d$ and is given in Equation 4.

The second factor in Equation 6 is the probability of observing $\boldsymbol{w}_{d,t}$ and $y_d$ given the new path assignments, $P(\boldsymbol{w}_{d,t}, y_d \,|\, c_{d,t} = c, \boldsymbol{w}^{-d,t}, \boldsymbol{c}^{-d,t}, \boldsymbol{t}, \boldsymbol{z}, \boldsymbol{\eta}) = f_c^{-d,t}(\boldsymbol{w}_{d,t}) \cdot g_c^{-d,t}(y_d)$.

**Optimizing $\eta$ and $\tau$:** We optimize the regression parameters $\eta$ and $\tau$ via the likelihood,

$$\mathcal{L}(\boldsymbol{\eta}, \boldsymbol{\tau}) = -\frac{1}{2\rho} \sum_{d=1}^{D} (y_d - \boldsymbol{\eta}^T \bar{\boldsymbol{z}}_d - \boldsymbol{\tau}^T \bar{\boldsymbol{w}}_d)^2 - \frac{1}{2\sigma} \sum_{k=1}^{K^+} (\eta_k - \mu)^2 - \frac{1}{\omega} \sum_{v=1}^{V} |\tau_v|, \qquad (7)$$

where $K^+$ is the number of nodes in the tree.[2] This maximization is performed using L-BFGS [13].

## 4  Data: Congress, Products, Films

We conduct our experiments using three datasets: Congressional floor debates, Amazon product reviews, and movie reviews. For all datasets, we remove stopwords, add bigrams to the vocabulary, and filter the vocabulary using tf-idf.[3]

- **U.S Congressional floor debates**: We downloaded debates of the $109^{th}$ US Congress from Gov-Track[4] and preprocessed them as in Thomas et al. [14]. To remove uninterestingly non-polarized debates, we ignore bills with less than 20% "Yea" votes or less than 20% "Nay" votes. Each document $d$ is a *turn* (a continuous utterance by a single speaker, i.e. *speech segment* [14]), and its response variable $y_d$ is the first dimension of the speaker's DW-NOMINATE score [15], which captures the traditional left-right political distinction.[5] After processing, our corpus contains 5,201 turns in the House, 3,060 turns in the Senate, and 5,000 words in the vocabulary.[6]
- **Amazon product reviews**: From a set of Amazon reviews of manufactured products such as computers, MP3 players, GPS devices, etc. [16], we focused on the 50 most frequently reviewed products. After filtering, this corpus contains 37,191 reviews with a vocabulary of 5,000 words. We use the rating associated with each review as the response variable $y_d$.[7]
- **Movie reviews**: Our third corpus is a set of 5,006 reviews of movies [17], again using review ratings as the response variable $y_d$, although in this corpus the ratings are normalized to the range from 0 to 1. After preprocessing, the vocabulary contains 5,000 words.

## 5  Evaluating Prediction

SHLDA's response variable predictions provide a formally rigorous way to assess whether it is an improvement over prior methods. We evaluate effectiveness in predicting values of the response variables for unseen documents in the three datasets. For comparison we consider these baselines:

- Multiple linear regression (MLR) models the response variable as a linear function of multiple features (or regressors). Here, we consider two types of features: topic-based features and lexically-based features. Topic-based MLR, denoted by MLR-LDA, uses the topic distributions learned by vanilla LDA as features [12], while lexically-based MLR, denoted by MLR-VOC, uses the frequencies of words in the vocabulary as features. MLR-LDA-VOC uses both features.
- Support vector regression (SVM) is a discriminative method [18] that uses LDA topic distributions (SVM-LDA), word frequencies (SVM-VOC), and both (SVM-LDA-VOC) as features.[8]
- Supervised topic model (SLDA): we implemented SLDA using Gibbs sampling. The version of SLDA we use is slightly different from the original SLDA described in [12], in that we place a Gaussian prior $\mathcal{N}(0,1)$ over the regression parameters to perform L2-norm regularization.[9]

For parametric models (LDA and SLDA), which require the number of topics $K$ to be specified before-hand, we use $K \in \{10, 30, 50\}$. We use symmetric Dirichlet priors in both LDA and SLDA, initialize

| Models | Floor Debates | | | | Amazon Reviews | | Movie Reviews | |
|---|---|---|---|---|---|---|---|---|
| | House-Senate | | Senate-House | | | | | |
| | PCC ↑ | MSE ↓ | PCC ↑ | MSE ↓ | PCC ↑ | MSE ↓ | PCC ↑ | MSE ↓ |
| SVM-LDA$_{10}$ | 0.173 | 0.861 | 0.08 | 1.247 | 0.157 | 1.241 | 0.327 | 0.970 |
| SVM-LDA$_{30}$ | 0.172 | 0.840 | 0.155 | 1.183 | 0.277 | 1.091 | 0.365 | 0.938 |
| SVM-LDA$_{50}$ | 0.169 | 0.832 | 0.215 | 1.135 | 0.245 | 1.130 | 0.395 | 0.906 |
| SVM-VOC | 0.336 | 1.549 | 0.131 | 1.467 | 0.373 | 0.972 | 0.584 | 0.681 |
| SVM-LDA-VOC | 0.256 | 0.784 | 0.246 | 1.101 | 0.371 | 0.965 | 0.585 | 0.678 |
| MLR-LDA$_{10}$ | 0.163 | 0.735 | 0.068 | 1.151 | 0.143 | 1.034 | 0.328 | 0.957 |
| MLR-LDA$_{30}$ | 0.160 | 0.737 | 0.162 | 1.125 | 0.258 | 1.065 | 0.367 | 0.936 |
| MLR-LDA$_{50}$ | 0.150 | 0.741 | 0.248 | 1.081 | 0.234 | 1.114 | 0.389 | 0.914 |
| MLR-VOC | 0.322 | 0.889 | 0.191 | 1.124 | 0.408 | 0.869 | 0.568 | 0.721 |
| MLR-LDA-VOC | 0.319 | 0.873 | 0.194 | 1.120 | 0.410 | **0.860** | 0.581 | 0.702 |
| SLDA$_{10}$ | 0.154 | **0.729** | 0.090 | 1.145 | 0.270 | 1.113 | 0.383 | 0.953 |
| SLDA$_{30}$ | 0.174 | 0.793 | 0.128 | 1.188 | 0.357 | 1.146 | 0.433 | 0.852 |
| SLDA$_{50}$ | 0.254 | 0.897 | 0.245 | 1.184 | 0.241 | 1.939 | 0.503 | 0.772 |
| SHLDA | **0.356** | 0.753 | **0.303** | **1.076** | **0.413** | 0.891 | **0.597** | **0.673** |

Table 2: Regression results for Pearson's correlation coefficient (PCC, higher is better (↑)) and mean squared error (MSE, lower is better (↓)). Results on Amazon product reviews and movie reviews are averaged over 5 folds. Subscripts denote the number of topics for parametric models. For SVM-LDA-VOC and MLR-LDA-VOC, only best results across $K \in \{10, 30, 50\}$ are reported. Best results are in **bold**.

the Dirichlet hyperparameters to 0.5, and use slice sampling [19] for updating hyperparameters. For SLDA, the variance of the regression is set to 0.5. For SHLDA, we use trees with maximum depth of three. We slice sample $m$, $\pi$, $\beta$ and $\gamma$, and fix $\mu = 0$, $\sigma = 0.5$, $\omega = 0.5$ and $\rho = 0.5$. We found that the following set of initial hyperparameters works reasonably well for all the datasets in our experiments: $m = 0.5$, $\pi = 100$, $\vec{\beta} = (1.0, 0.5, 0.25)$, $\vec{\gamma} = (1, 1)$, $\alpha = 1$. We also set the regression parameter of the root node to zero, which speeds inference (since it is associated with every document) and because it is reasonable to assume that it would not change the response variable.

To compare the performance of different methods, we compute Pearson's correlation coefficient (PCC) and mean squared error (MSE) between the true and predicted values of the response variables and average over 5 folds. For the Congressional debate corpus, following Yu et al. [20], we use documents in the House to train and test on documents in the Senate and vice versa.

**Results and analysis**    Table 2 shows the performance of all models on our three datasets. Methods that only use topic-based features such as SVM-LDA and MLR-LDA do poorly. Methods only based on lexical features like SVM-VOC and MLR-VOC outperform methods that are based only on topic features significantly for the two review datasets, but are comparable or worse on congressional debates. This suggests that reviews have more highly discriminative words than political speeches (Table 3). Combining topic-based and lexically-based features improves performance, which supports our choice of incorporating both per-topic and per-word regression parameters in SHLDA.

In all cases, SHLDA achieves strong performance results. For the two cases where SHLDA was second best in MSE score (Amazon reviews and House-Senate), it outperforms other methods in PCC. Doing well in PCC for these two datasets is important since achieving low MSE is relatively easier due to the response variables' bimodal distribution in the floor debates and positively-skewed distribution in Amazon reviews. For the floor debate dataset, the results of the House-Senate experiment are generally better than those of the Senate-House experiment, which is consistent with previous results [20] and is explained by the greater number of debates in the House.

## 6    Qualitative Analysis: Agendas and Framing/Perspective

Although a formal coherence evaluation [21] remains a goal for future work, a qualitative look at the topic hierarchy uncovered by the model suggests that it is indeed capturing agenda/framing structure as discussed in Section 1. In Figure 3, a portion of the topic hierarchy induced from the Congressional debate corpus, Nodes A and B illustrate agendas—issues introduced into political discourse—associated with a particular ideology: Node A focuses on the hardships of the poorer victims of hurricane Katrina and is associated with Democrats, and text associated with Node E discusses a proposed constitutional amendment to ban flag burning and is associated with Republicans. Nodes C and D, children of a neutral "tax" topic, reveal how parties frame taxes as *gains* in terms of new social services (Democrats) and *losses* for job creators (Republicans).

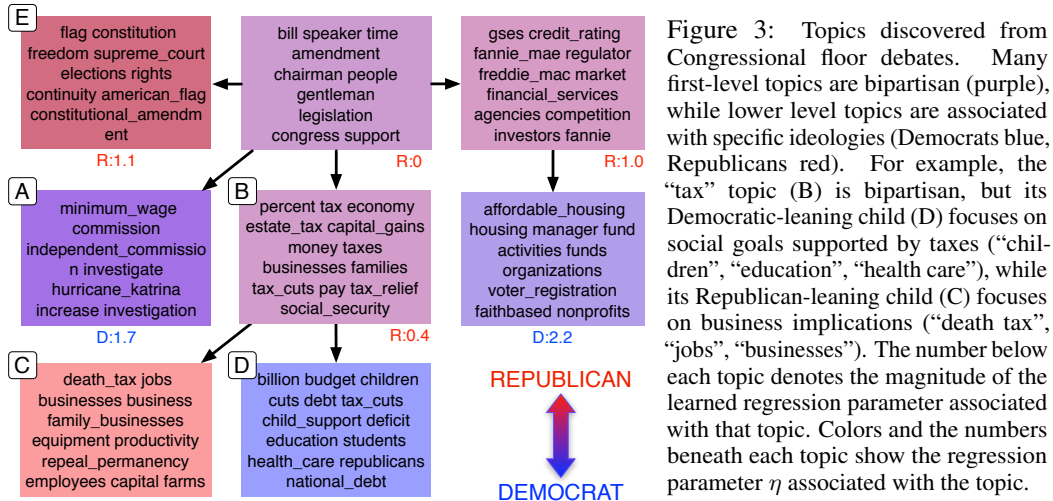

Figure 3: Topics discovered from Congressional floor debates. Many first-level topics are bipartisan (purple), while lower level topics are associated with specific ideologies (Democrats blue, Republicans red). For example, the "tax" topic (B) is bipartisan, but its Democratic-leaning child (D) focuses on social goals supported by taxes ("children", "education", "health care"), while its Republican-leaning child (C) focuses on business implications ("death tax", "jobs", "businesses"). The number below each topic denotes the magnitude of the learned regression parameter associated with that topic. Colors and the numbers beneath each topic show the regression parameter $\eta$ associated with the topic.

Figure 4 shows the topic structure discovered by SHLDA in the review corpus. Nodes at higher levels are relatively neutral, with relatively small regression parameters.[10] These nodes have general topics with no specific polarity. However, the bottom level clearly illustrates polarized positive/negative perspective. For example, Node A concerns washbasins for infants, and has two polarized children nodes: reviewers take a positive perspective when their children enjoy the product (Node B: "loves", "splash", "play") but have negative reactions when it leaks (Node C: "leak(s/ed/ing)").

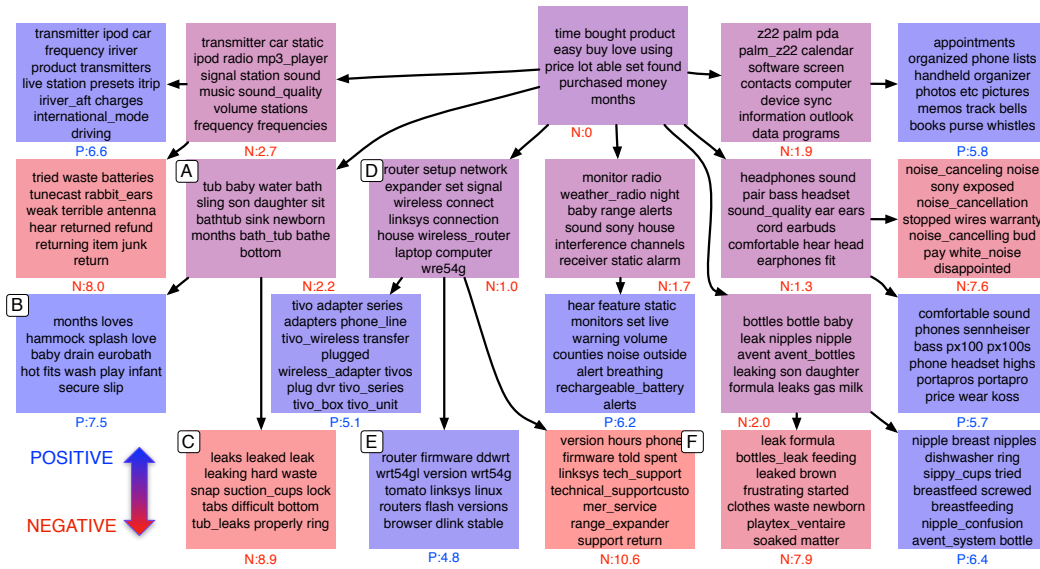

Figure 4: Topics discovered from Amazon reviews. Higher topics are general, while lower topics are more specific. The polarity of the review is encoded in the color: red (negative) to blue (positive). Many of the first-level topics have no specific polarity and are associated with a broad class of products such as "routers" (Node D). However, the lowest topics in the hierarchy are often polarized; one child topic of "router" focuses on upgradable firmware such as "tomato" and "ddwrt" (Node E, positive) while another focuses on poor "tech_support" and "customer_service" (Node F, negative). The number below each topic is the regression parameter learned with that topic.

In addition to the per-topic regression parameters, SHLDA also associates each word with a lexical regression parameter $\tau$. Table 3 shows the top ten words with highest and lowest $\tau$. The results are unsuprising, although the lexical regression for the Congressional debates is less clear-cut than other

datasets. As we saw in Section 5, for similar datasets, SHLDA's context-specific regression is more useful when global lexical weights do not readily differentiate documents.

| Dataset | Top 10 words with positive weights | Top 10 words with negative weights |
|---|---|---|
| Floor Debates | bringing, private_property, illegally, tax_relief, regulation, mandates, constitutional, committee_report, illegal_alien | bush_administration, strong_opposition, ranking, republicans, republican_leadership, secret, discriminate, majority, undermine |
| Amazon Reviews | highly_recommend, pleased, love, loves, perfect, easy, excellent, amazing, glad, happy | waste, returned, return, stopped, leak, junk, useless, returning, refund, terrible |
| Movie Reviews | hilarious, fast, schindler, excellent, motion_pictures, academy_award, perfect, journey, fortunately, ability | bad, unfortunately, supposed, waste, mess, worst, acceptable, awful, suppose, boring |

Table 3: Top words based on the global lexical regression coefficient, $\tau$. For the floor debates, positive $\tau$'s are Republican-leaning while negative $\tau$'s are Democrat-leaning.

# 7 Related Work

SHLDA joins a family of LDA extensions that introduce hierarchical topics, supervision, or both. Owing to limited space, we focus here on related work that combines the two. Petinot et al. [22] propose hierarchical Labeled LDA (hLLDA), which leverages an observed document ontology to learn topics in a tree structure; however, hLLDA assumes that the underlying tree structure is known *a priori*. SSHLDA [23] generalizes hLLDA by allowing the document hierarchy labels to be partially observed, with unobserved labels and topic tree structure then inferred from the data. Boyd-Graber and Resnik [24] used hierarchical distributions *within* topics to learn topics across languages. In addition to these "upstream" models [25], Perotte et al. [26] propose a "downstream" model called HSLDA, which jointly models documents' hierarchy of labels and topics. HSLDA's topic structure is flat, however, and the response variable is a hierarchy of labels associated with each document, unlike SHLDA's continuous response variable. Finally, another body related body of work includes models that jointly capture topics and other facets such as ideologies/perspectives [27, 28] and sentiments/opinions [29], albeit with discrete rather than continuously valued responses.

Computational modeling of sentiment polarity is a voluminous field [30], and many computational political science models describe agendas [5] and ideology [31]. Looking at framing or bias at the sentence level, Greene and Resnik [32] investigate the role of syntactic structure in framing, Yano et al. [33] look at lexical indications of sentence-level bias, and Recasens et al. [34] develop linguistically informed sentence-level features for identifying bias-inducing words.

# 8 Conclusion

We have introduced SHLDA, a model that associates a continuously valued response variable with hierarchical topics to capture both the issues under discussion and alternative perspectives on those issues. The two-level structure improves predictive performance over existing models on multiple datasets, while also adding potentially insightful hierarchical structure to the topic analysis. Based on a preliminary qualitative analysis, the topic hierarchy exposed by the model plausibly captures the idea of agenda setting, which is related to the issues that get discussed, and framing, which is related to authors' perspectives on those issues. We plan to analyze the topic structure produced by SHLDA with political science collaborators and more generally to study how SHLDA and related models can help analyze and discover useful insights from political discourse.

# Acknowledgments

This research was supported in part by NSF under grant #1211153 (Resnik) and #1018625 (Boyd-Graber and Resnik). Any opinions, findings, conclusions, or recommendations expressed here are those of the authors and do not necessarily reflect the view of the sponsor.

## Footnotes

[1]We emphasize that, unlike in HDP where each table is assigned to a single dish, each *table* in our metaphor is associated with a combo–a collection of $L$ dishes. We also use *combo* and *path* interchangeably.

[2]The superscript $^+$ is to denote that this number is unbounded and varies during the sampling process.

[3]To find bigrams, we begin with bigram candidates that occur at least 10 times in the corpus and use Pearson's $\chi^2$-test to filter out those that have $\chi^2$-value less than 5, which corresponds to a significance level of 0.025. We then treat selected bigrams as single word types and add them to the vocabulary.

[4] http://www.govtrack.us/data/us/109/

[5]Scores were downloaded from http://voteview.com/dwnomin_joint_house_and_senate.htm

[6]Data will be available after blind review.

[7]The ratings can range from 1 to 5, but skew positive.

[8] http://svmlight.joachims.org/

[9]This performs better than unregularized SLDA in our experiments.

[10]All of the nodes at the second level have slightly negative values for the regression parameters mainly due to the very skewed distribution of the review ratings in Amazon.

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
