[Reviews · NeurIPS 2013]

Submitted by Assigned_Reviewer_5

This paper proposes a hierarchical, supervised topic model that builds on
ideas from hierarchical LDA using the nested CRP and from supervised LDA,
which associates topics with regression coefficients. The proposed model
further extends hierarchical LDA by allowing each sentence in a document to
chose its own path through the topic tree, and sharing of paths for sentences
within the same document is modelled through a document-level CRP where the
customers are sentences and the dishes are paths through the topic hierarchy
("combos"). Another difference to supervised LDA is that each word is also
associated with its own regression coefficient to model topic-independent
effects. The authors empirically demonstrate the performance of the model
on 3 data sets and show that the proposed model outperforms LDA+linear regression
as well as supervised LDA.

The idea of modelling framing in the proposed way by associating regression
coefficients with hierarchically arranged topics is interesting and appears
novel. The paper is clearly written and the proposed model and inference procedure
seem technically sound. The quantitative evaluation shows that the proposed model
(slightly) outperforms simple baselines and the qualitative evaluation suggests that
the discovered topic hierarchy has the desired property of separating topics into different
aspects or ways of framing them.

The main question I had while reading this paper was: Is such a complicated
model really necessary for modelling this what/how distinction? Given that the
depth of the hierarchy is fixed to 3 in the experiments (why?), it seems
that at least an infinitely deep hierarchy is not necessary. If interpretability
of the resulting topics is desired, would it make sense to model the binary
split (positive/negative, republican/democrat) explicitly, e.g. by explicitly
splitting each topic into pos/neutral/neg parts with fixed regression coefficients
and then associating with each document a distribution over those parts?

In terms of quantitative experiments, it would be nice to tease apart the contributions
of the individual changes to existing models: how does sLDA perform if lexical
regression parameters are added, or a supervised response variable is added
to a plain HLDA model? Also, it would be helpful to see some descriptive
statistics about the inference and the learned models, e.g. how many nodes are
in the hierarchy, branching factor, iterations until convergence, etc.
Summary: The paper proposes a novel extension of a combination of hierarchical and supervised LDA that can hierarchically split topics into subtopic depending on their correlation with the response variable. The paper is well executed, but on the other hand it's a very complicated model that yields only marginal quantitative improvements over much simpler methods.

Submitted by Assigned_Reviewer_6

Quality:
This is a well constructed model. The algorithm uses the state of the art techniques here, copying previously published methods. The key idea, however, is the idea of dropping HLDA's overly constrained "path per document" formulation to make a "path per sentence" formulation. A related idea of allowing topics to have a higher fidelity in parts of documents (combining sentence/segment-based and document-based effects, e.g., Ponti, Tagarelli, and Karypis in DS 2011 are one of many doing this) is well established for LDA, and using hLDA for this is easier due to its restricted form. Another good idea is adding words to the regression, although this has been done with LDA-augmented use of SVMs on text for a decade.
The introductory discussion, agenda setting and framing, is unfortunately not properly evaluated later on in the paper. Its also not clear whether Amazon reviews should be included in this discussion.
Top of page 6 you say you slice sample some parameters and then immediately after you say you fix them to preset values. I am confused: which do you do?
Section 6 you give a qualitative analysis. This is unsatisfactory. While these all look good and make your case, too much is left out. How many topics where uncovered, or whats the effective number of topics? So what fraction are you showing? Coherence evaluation is easy to do. There isn't much excuse for not doing it. At least you could have got some evaluators to blindly assess topics and give some opinions. How many topics where "junk" and uninterpretable? How many made sense to well-educated political watchers? Again, this is easy to set up and can be done in a blind test (randomly assign individual topics).

Clarity: Well written.

Other: "china" in refs needs a "C"; figures 3 and 4 are too small, needed 150% display to read text; footnote 3 attached at the wrong place, its not about tf-idf.

Originality: the key innovation is the combination of modelling constructs for the particular task. This is, I believe, a good illustration of where we are now with a lot of topic modelling, and its done well in this paper.

Significance: nicely presented results with a simple, well explained model and straight forward algorithm. Thus should have some impact because its easy to "get" and the results are good. Release the code and we will see!
Summary: Nice prediction results with poor qualitative evaluation on a well explained model and straight forward algorithm. Mature presentation, though some adjustments needed.

Submitted by Assigned_Reviewer_7

This paper propose a supervised hierarchical topic model for analyzing both topical and ideological perspectives in text documents with response variables. It is interesting to obtain a hierarchical structure, in which higher level nodes show topics and lower level nodes show ideologies. The experiment is intensive; the authors demonstrate high predictive performance of the proposed model in three data sets by comparing many other methods.

The proposed model is a bit complicated. Why multiple paths for each document is required for this task? A simple combination of nested CRP and supervised topic model might be able to obtain a similar hierarchical structure, where both agenda and framing are represented. I would like to see discussion on this.

Does the proposed model improve perplexity comparing other related models, such as LDA and nested HDP, with/without response variables?

The inference seems to take time. The information about computational time of the proposed model would be informative for readers.
Summary: The proposed model is a combination of nested Chinese restaurant process and supervised topic models. The experiments are extensive.
Author Feedback

Author rebuttal: We thank the reviewers for the care they’ve taken with this review and for their comments and suggestions.

=== The Complexity of the Model and Alternative Assumptions ===

Reviewer_5 and Reviewer_7 have different comments on the complexity of the proposed model relative to the task.

Reviewer_7 suggests a simpler version of our proposed model: using nCRP with a supervised topic model. This, too, was our initial idea, but it was often worse than SLDA. The main problem with this is that since each path represents a consistent theme (with different levels of abstraction represented by different nodes on the path), using a single path to model a document, especially a long document, is unsatisfactory. In this regard, the way our model improves over Reviewer_7’s suggestion is similar to the way LDA improves over a mixture model.

Reviewer_5 suggests a less complicated, fixed 3-level hierarchy with topics at the 2nd level and a split between positive/neutral/negative or the ideological dimension at the leaf node level. This makes sense for the present application, but even though our immediate motivation is to capture framing, where specifically a three-level hierarchy is motivated by theoretical treatments of framing as second-level agenda-setting, we also would like to take a general approach by proposing a model that is capable of capturing flexible hierarchies. In our experiments, we followed previous work (e.g., Blei et. al. JACM 2010) to fix the tree heights to 3 to facilitate interpretation/visualization of the results. This also makes the learning time more manageable. However, we believe that with its flexible topic structure, our proposed model is applicable to many other domains. We will note this in our final version.

It is also worth noting that our more general approach discovers interesting second-level effects beyond merely identifying the (sub)topics. For instance, in the Congressional data our model only discovers a second-level topic about the flag burning amendment; this does not have lower-level specialized topics (e.g. demonstrating a Republican / Democrat distinction). The regression parameter associated with this topic is Republican, however, suggesting that rather than there being well defined Republican / Democrat framings on this topic, merely talking about a flag-burning amendment is associated with taking a Republican position.

=== Qualitative / Topic Coherence Evaluation ===

Reviewer_6 expresses some concerns about the qualitative evaluation and Reviewer_7 suggests using perplexity to evaluate the topic quality.

We have conducted some preliminary experiments on this, but we believe that evaluating the coherence of topics learned by hierarchical / nonparametric models is a research area in its own right. As suggested in previous work (Chang et. al. NIPS 2009), perplexity is not the best metric to capture topic coherence. Therefore, we turned our focus on a recent topic coherence metric proposed by Mimno et. al. (EMNLP 2011). However, this metric is not very suitable for evaluating a hierarchy of topics since it favors “big” topics (i.e., topics with many tokens assigned via Gibbs sampling). In our case, “small” topics which explain just a few tokens are common and will likely to be “incoherent” according to Mimno et. al.’s metric.

Also, as mentioned in the paper, we have active collaborations with social scientists to better examine the induced latent space. We plan to take this up in greater and more rigorous detail in future work.

=== Hyperparameter Optimization ===

For Reviewer_6’s question about the way we performed hyperparameter optimization: we performed slice sampling to optimize hyperparameters; the values described in Section 6 are the *initial* values we set before slice sampling was performed. We apologize for the confusion and will emphasize that these are initial values.